SUMOylation in Trypanosoma brucei

Klein Cornelia Andrea
Droll Dorothea
Clayton Christine cclayton@zmbh.uni-heidelberg.de
Zentrum für Molekulare Biologie der Universität Heidelberg, DKFZ-ZMBH Alliance , Heidelberg , Germany
Moreno Silvia
Electronic publication date: 2013 Oct 8
Publication date: 2013
Volume: 1
Electronic Location ID: e180
Received 2012 Dec 4; Accepted 2013 Sep 18
Copyright: © 2013 Klein et al.
Copyright year: 2013
Copyright holder: Klein et al.
License: This is an open access article distributed under the terms of the Creative Commons Attribution License, which permits unrestricted use, distribution, and reproduction in any medium, provided the original author and source are credited.
License URL: https://creativecommons.org/licenses/by/3.0/

Keywords: Trypanosoma brucei, SUMO, Heat shock, Oxidative stress, Small ubiquitin-like modifier

Funding: Sonderforschungsbereich 544 of the Deutsche Forschungsgemeinschaft DD was supported by Sonderforschungsbereich 544 of the Deutsche Forschungsgemeinschaft. The funder had no role in study design, data collection and analysis, decision to publish, or preparation of the manuscript.

==============================
Small ubiquitin like modifier (SUMO) proteins are involved in many processes in eukaryotes. We here show that Trypanosoma brucei SUMO (Tb927.5.3210) modifies many proteins. The levels of SUMOylation were unaffected by temperature changes but were increased by severe oxidative stress. We obtained evidence that trypanosome homologues of the SUMO conjugating enzyme Ubc9 (Tb927.2.2460) and the SUMO-specific protease SENP (Tb927.9.2220) are involved in SUMOylation and SUMO removal, respectively.

Introduction

Small ubiquitin like modifier (SUMO) proteins have been found in almost all eukaryotes. Conjugation of SUMO to target proteins alters their functions in multiple ways, and it is therefore central to a multitude of different cellular processes.

Like ubiquitin, SUMO is attached to its targets via 3 enzymatic steps (Geiss-Friedlander & Melchior, 2007; Ulrich, 2009). First, a SUMO-specific protease (SENP) removes 2–11 amino acids at the SUMO C-terminus, revealing a C-terminal di-glycine motif (Kim & Baek, 2009). Next, SUMO is activated by the SUMO activation complex (E1 complex), which consists of two enzymes, Aos1/SAE1 (budding yeast/human) and Uba2/SAE2 (Johnson, 1997; Desterro, 1999). The C-terminal glycine of SUMO forms a thioester bond with a cysteine residue of Uba2. From there, it is transferred to a cysteine residue of the E2 SUMO conjugating enzyme (Ubc9) (Johnson & Blobel, 1997; Desterro, Thomson & Hay, 1997). From the E2 conjugation enzyme, SUMO binds to a target lysine residue (Geiss-Friedlander & Melchior, 2007; Ulrich, 2009). This process is assisted by an E3 ligase. SUMO is usually attached as a monomer, although chain formation can occur (Ulrich, 2008). SUMO is removed from its targets by a variety of peptidases called SENPs (Mukhopadhyay & Dasso, 2007). SENP regulation is critical for homeostasis (Kim & Baek, 2009; Xu et al., 2009; Yeh, 2009; Drag & Salvesen, 2008) and is also involved in responses to stresses such as heat shock and oxidation (Xu et al., 2009; Tempé, Piechaczyk & Bossis, 2008). SUMO is essential for growth in S. cerevisiae (Johnson, 1997) but not in fission yeast (Tanaka et al., 1999) or Aspergillus (Szewczyk et al., 2008). Work on Chlamydomonas revealed that the abundance of SUMOylated proteins increases during heat shock and osmotic stress (Wang et al., 2008). SUMO was also examined in Toxoplasma gondii (Braun et al., 2009) and Plasmodium falciparum (Issar et al., 2008): in both cases many SUMOylated proteins were observed and identified by mass spectrometry, but details of the roles of SUMO in specific processes are not yet known.

In the kinetoplastid Trypanosoma cruzi, the components of the SUMOylation machinery have been identified by BLAST search. Numerous SUMOylated bands were identified by Western blotting using both anti-SUMO antibody and detection of epitope-tagged SUMO. In addition, 236 potentially SUMOylated proteins were identified by tandem affinity purification and mass spectrometry (Bayona et al., 2011), but unfortunately, a recent careful re-examination of the spectra could unambiguously identify only eight SUMOylated peptides on just seven proteins (Xu et al., 2013). One of the targets identified by the tandem affinity purification, metacaspase 3, was confirmed by co-immunoprecipitation (Bayona et al., 2011) although the SUMOylated peptide was not found (Xu et al., 2013). The paraflagellar rod protein PFR1 (also called PAR3) was suggested as a SUMO target by Western blot analysis and in vitro SUMOylation (Annoura et al., 2012) but again no SUMOylated peptide was found (Xu et al., 2013). T. cruzi SUMO itself has a SUMOylation site and is able to polymerize (Annoura et al., 2012). Together these results suggest that many proteins are SUMOylated in T. cruzi, but purification is very difficult. Possibly, SUMO protease is very active and persists during purification procedures.

The amino-acid sequence of Trypanosoma brucei SUMO (TbSUMO, Tb927.5.3210) is 37% identical with that of human SUMO-1 and the 3D structure (solved using NMR) is similar to those of yeast and mammalian SUMO (Shang et al., 2009). It was shown by chemical shift analysis that TbSUMO interacts with human Ubc9. RNAi targeting SUMO in procyclic trypanosomes caused growth arrest and cell death, and HA-tagged SUMO was predominantly found in the T. brucei nucleus (Liao et al., 2010). RNA interference targeting SUMO in T. brucei caused growth arrest, followed by death, and antibodies to SUMO in bloodstream-form parasites recognised two prominent bands at around 55 and 60 kDa (Obado et al., 2011).

In this paper we describe preliminary functional characterisation of further components of the SUMOylation system in T. brucei and investigate the effects of various stresses on protein SUMOylation.

Methods

Plasmids

For the N-terminal in situ TAP tag, a part of the TbSUMO open reading frame (ORF) was amplified using the following primers fw: 5′-gac aag ctt ccg cca ccg acg aac cca ctc ata ac-3′ rv: 5′-gtc gat atc tca tgt ctg ctc cac cat cgc-3′ and cloned into the p2676 vector (Kelly et al., 2007) using Hind III and EcoR V.

For the N-terminal V5 in situ tag, a part of the TbSUMO ORF was amplified (fw: 5′- gac ctc gag gac gaa ccc act cat-3′, rv; 5′- gac ggg ccc tca cgc cat gca cca-3′), as well as a part of the 5′ untranslated region (UTR) (fw: 5′- gac ccg cgg tgt cct tgt ggt tac gt-3′ rv: 5′-gac tct aga aag agg aag tcg ggg ag-3′). The ORF and UTR fragments were cloned into a vector containing the V5-tag and the Blasticidin resistance as described in Shen et al. (2001) using Apa I and Xho I for the ORF and Sac II and Xba I for the UTR fragment.

For the RNAi constructs, portions of the open reading frames of the targeted genes were amplified and cloned into p2T7TA blue (Alibu et al., 2004). The following primers were used: for TbSUMO fw: 5′-ggg ggt acc gac gaa ccc act cat aac-3′ rv: 5′-ccc aag ctt tca cgc cat gca cca aag 3′; for Tb09.160.0970 (TbSENP) fw: 5′-cag acg act cac tat cgc ca-3′, rv:5′-tgc gct caa atg ttg ttc tc-3′ and for Tb927.2.2460 (TbUBC9) fw: 5′-tag ctc agt cac gcc tac ga-3′ rv:5′-aca cac gaa atg gct ctt cc-3′. The primers were designed using RNAit (Redmond, Vadivelu & Field, 2003).

Trypanosome culture

Trypanosoma brucei strain Lister 427 expressing the tet repressor, with or without T7 polymerase, were used throughout (Alibu et al., 2004), with culturing and transfecting of trypanosomes as previously described (van Deursen et al., 2001).

For growth studies, bloodstream-form cells were diluted to a starting concentration of 5 × 104 cells/ml, with a maximum density of 1.5–2 × 106 cells/ml. Procyclics were diluted to 5 × 105, with a maximum density of 5 × 106. Tetracycline was added to a final concentration of 0.25 µg/ml to induce expression from tetracycline-regulated promoters.

For differentiation, bloodstream-form cells were grown to 1.5–2 × 106 cells/ml, then cis-aconitate was added to a concentration of 6 mM. The cells were grown for 16 h at 37°C then transferred to 27°C. Inhibition of glucose transport was achieved by adding phloretin to a concentration of 100 µM. For oxidative stress, H2O2 was added to procyclic trypanosomes to final concentrations of 250 µM, 125 µM, 62.5 µM, 31.25 µM or 15.6 µM; the cells were harvested after an incubation time of 1 h.

Tandem affinity purification

For each tandem affinity purification 4–5 × 109 cells were harvested at 4°C and washed twice with ice-cold PBS containing 20 mM N-ethyl maleimide (NEM). Bloodstream cells were harvested at a density of 2 × 106 cells/ml, procyclic cells at a density of 5 × 106 cells/ml. The cell pellets were snap-frozen in liquid nitrogen and stored at −80°C. Cell breakage was performed in 6 ml breakage buffer (10 mM Tris-Cl, 10 mM NaCl, 0.1% NP40, 20 mM NEM, one tablet of complete inhibitor (without EDTA, Roche) pH = 7.8) by passing the cells 20–25 times through a 21 gauge needle. The lysate was spun at 13,000 g for 30 min at 4°C to remove the cell debris. Then NaCl was added to a final concentration of 150 mM. The purification was done according to Puig et al. (2001). 20 mM NEM was added to all the buffers, except during the wash and elution step of the IgG beads and during the TEV cleavage, as NEM inhibits TEV protease.

V5 immunoprecipitation

2.5–5 × 109 procyclic cells were harvested at a density of approximately 5 × 106 cells/ml, washed twice with ice-cold PBS containing 10 mM NEM and 10 mM iodoacetamide (IAA) and snap-frozen in liquid nitrogen and stored at −80°C. For use, the cell pellet was resuspended in 1 ml lysis buffer (10 mM Tris-C, 10 mM NaCl, 0.1% NP40, 1% SDS, complete protease inhibitor (Roche), 10 mM NEM, 10 mM IAA, pH = 7.5). Cells were passed the cells 20–25 times through a 21 gauge needle. The lysate was spun at 13,000 g for 30 min at 4°C to remove the cell debris and was diluted 1:10 in IP100 (10 mM Tris-Cl, 100 mM NaCl, 0.1% NP40, complete protease inhibitor (Roche), 10 mM NEM, 10 mM IAA).

Before immunoprecipitation, the lysate was incubated with protein A sepharose for 1 h on a rotary shaker at 4°C to absorb non-specifically binding proteins. The supernatant was then added to 50 µl α-V5 beads (self-made or from Sigma). The lysate was incubated for 3 h on a rotary shaker at 4°C. The beads were washed seven times with IP 100. Elution was done by incubating the beads twice with 125 µl IP buffer mixed with 25 µl V5 peptide for 30 min, then boiling in 4 × Laemmli buffer.

Results and Discussion

Many proteins are SUMOylated in T. brucei

To detect SUMOylated proteins, an antibody was raised to His-tagged TbSUMO produced in E. coli. (For details of all plasmid constructs see Table 1.) The anti-T. brucei SUMO antibody was unfortunately insufficiently specific. Although it recognised purified recombinant SUMO, it detected several bands, but not monomeric SUMO, in bloodstream- and procyclic-form cell extracts and the banding pattern was not affected by SUMO RNAi.

Table 1 Plasmids used in this work (not all results described in text).

Plasmid	Description	Cloning strategy	
pHD2020	SUMO/TAP in-situ tag
(N-terminal)	A part of the SUMO ORF was amplified using the following primers
fw: 5′-gac aag ctt ccg cca ccg acg aac cca ctc ata ac-3′
rv: 5′-gtc gat atc tca tgt ctg ctc cac cat cgc-3′
and cloned into the p2676 (Kelly et al., 2007) using Hind III and Eco RV	
pHD2021	SUMO/V5 in-situ tag
(N-terminal)	The SUMO ORF was amplified using the following primers:
fw: 5′-gac ctc gag gac gaa ccc act cat-3′
rv: 5′-gac ggg ccc tca cgc cat gca cca-3′.
A fragment of the UTR was amplified using the following primers:
fw: 5′-gac ccg cgg tgt cct tgt ggt tac gt-3′
rv: 5′-gac tct aga aag agg aag tcg ggg ag-3′.
The ORF was cloned into the Bla V5 vector (Shen et al., 2001) using Apa I and
Xho I, the 5′ UTR using Sac II and Xba I.	
pHD2022	SUMO knock-out	A fragment of the SUMO 3′ UTR was amplified using the following primers:
fw: 5′-gac tct aga cat aag tgc gcg tag tgg-3′
rv: 5′-gtc ccg cgg gca aac gac cgc aga agt-3′.
A 5′-UTR fragment was amplified using the following primers:
fw: 5′-cac tcg agc cct cat atc cac atc ctc a-3′
rv: 5′- gtc aag ctt cgt ggg ctc aga aat gaa-3′.
The 3′-UTR fragment was cloned into pHD1748 (Blasticidin resistance cassette in polylinker) using
Xba I and Sac II, the 5′ UTR fragment using Xho I and Hind III.	
pHD2023	SUMO knock-out	The Blasticidin resistance was removed from pHD2022 using Hind III and Eco RI and replaced by a Puromycin resistance which was acquired by digesting pHD1747 with the same enzymes.	
pHD2024	SUMO RNAi	The SUMO ORF was amplified using the following primers:
fw: 5′-ggg ggt acc gac gaa ccc act cat aac-3′
rv: 5′-ccc aag ctt tca cgc cat gca cca aag -3′
and cloned into p2T7TA blue (Alibu et al., 2004)	
pHD2025	His-SUMO	The SUMO ORF was amplified using the following primers:
fw: 5′-gag ggt acc gac gaa ccc act cat aac-3′
rv: 5′-ccc aag ctt tca cgc cat gca cca aag-3′
and cloned into pQEA38 using Kpn I and Hind III. pQEA38 is an expression vector with ten His tags and a TEV cleavage site, modified from pQTEV (AY243506), from the lab of D Görlich (then at ZMBH).	
pHD2026	SUMO/His in-situ tag
(N-terminal)	A 5′ UTR fragment was amplified using the following primers:
fw: 5′-gac ccg cgg tgt cct tgt ggt tac gt-3′
rv: 5′-gac tct aga aag agg aag tcg ggg ag-3′
and cloned into the Bla V5 vector using Sac II and Xba I. Then the V5 tag was removed using Eco NI and Xho I.
The SUMO ORF and the His-tag sequence were cut out of pHD 2025 using Eco RI and Hind III.
The vector and the tagged ORF were blunted using the Klenow fragment and ligated.	
pHD2037	TbSENP RNAi	A fragment of the SENP ORF was amplified using the following primers:
fw: 5′-cag acg act cac tat cgc ca-3′
rv: 5′-tgc gct caa atg ttg ttc tc-3′
and cloned into the p2T7TA blue vector	
pHD2038	TbUBC9 RNAi	A fragment of the UBC9 ORF was amplified using the following primers
fw: 5′-tag ctc agt cac gcc tac ga-3′
rv: 5′-aca cac gaa atg gct ctt cc-3′
and cloned into the p2T7TA blue vector	

Next, in bloodstream-form trypanosomes, we integrated a sequence encoding a tandem affinity purification tag (TAP-tag) N-terminally in frame with one allele of SUMO. The TAP-SUMO was seen as a ∼40 kDa band (Fig. 1A); the expected size was 33.5 kDa, comprising 12.5 kDa SUMO + 21 kDa tag. Many SUMOylated proteins were present, with a prominent band at 100 kDa, which, without the tag, would correspond to an 80 kDa SUMOylated conjugate. In T. cruzi also, using either HA-tagged or untagged SUMO, many SUMOylated bands were seen in addition to monomeric SUMO (Bayona et al., 2011). The pattern that we observed, however, bears no resemblance to the two prominent bands at 55 and 60 kDa that were previously described for bloodstream-form T. brucei using an anti-SUMO antibody (Obado et al., 2011). There are several possible explanations for the discrepancy. Our TAP-SUMO may have impaired function due to the tag, or the 100 kDa band could actually be equivalent to the 55KDa band (but with very aberrant migration). Alternatively the previously-published pattern could have been affected by antibody cross-reactions with abundant proteins. Since, in the published results, the antibody staining was abolished after 72 h RNAi, the last possibility seems unlikely. It cannot, however, be ruled out since no evidence was presented that the antibody recognised native monomeric SUMO on the Western blots. Also the RNAi had depleted the SUMO RNA within 24 h but the signal on the blots was unchanged after 48 h RNAi despite extensive morphological degeneration (Obado et al., 2011).

Figure 1 Protein modification by tagged SUMO.

(A) Effect of temperature on the pattern of modification with TAP-SUMO in bloodstream trypanosomes containing the plasmid pHD2020. Lane 1: cells without TAP-SUMO. Lanes 2, 3, 4: The cells were incubated for 1 h at the indicated temperatures. The antibody used for detection was PAP: peroxidase anti-peroxidase antibody (binds to the IgG-binding domain of the TAP tag). (B) Effect of differentiation conditions on the pattern of TAP-SUMOylated proteins. Bloodstream trypanosomes were isolated at 6 × 105 cells/ml (L, lower density, lane 1) or 2.5 × 106 cells/ml (H, higher density, lane 2). Cis-aconitate was added to the higher-density cells and the culture incubated at 37°C for 17 h (lanes 3–6) (Queiroz et al., 2009). Then, the culture was centrifuged and resuspended in procyclic-form medium at 27°C (lanes 7 & 8). (C) Effect of oxidative stress and temperature stress on the pattern of V5-SUMO modification in procyclic trypanosomes. Parasites were transfected with pHD2021 to V5-in situ tag SUMO at the N-terminus (Shen et al., 2001). Lanes 1 and 7 are controls. Lanes 2–6: Cells with V5-in situ-tagged SUMO were incubated for 1 h with 15.6, 31.2, 62.5, 125 or 250 µM hydrogen peroxide. Lanes 8–10: incubation for 1 h at the indicated temperatures. Proteins were detected with anti-V5; as a control, a monoclonal antibody to tubulin (TUB) (from K Gull) was used.

Our pattern of TAP-SUMO-conjugated proteins was unaffected by heat shock (Fig. 1A) or by treatment for 12 h or 24 h with a sub-lethal level (100 µM) of phloretin (Haanstra et al., 2011) to partially inhibit glucose import (not shown). The bloodstream forms used for these experiments are not able to differentiate into growing procyclic forms, but can undergo some early steps of differentiation after addition of cis-aconitate and transfer to 27°C. When we did this in the TAP-SUMO-expressing line, the banding pattern remained largely unchanged, but one band migrating at 90 kDa reproducibly disappeared (Fig. 1B, marked with a star). In accordance with this result, a changing SUMOylation pattern during differentiation was found in T. cruzi (Annoura et al., 2012). Our experiments only detected the most abundant SUMOylated proteins and it is quite possible that less abundant proteins show regulated SUMOylation.

In procyclic forms (Fig. 1C), we integrated a sequence encoding a V5 epitope tag upstream of the open reading frame (Shen et al., 2001). We expected monomeric V5-SUMO to migrate at 13 kDa. This was not reproducibly seen, but we did sometimes see a band or bands running at 20 kDa (Fig. 1C). In contrast, slower mobility bands were always present, in particular a prominent double band just below 100 kDa. Comparison of the patterns from bloodstream and procyclic forms (by manipulating the photographs to allow for the sizes of the tags, not shown) suggested that the patterns of abundant SUMOylated proteins were similar in both forms. The SUMOylation pattern in procyclics was unaffected by temperature changes (1 h incubations, Fig. 1C lanes 7–10).

SUMOylation increases after oxidative stress

In mammalian cells, peroxide concentrations of 1 mM and lower inhibit SUMOylation (Bossis & Melchior, 2006) through formation of a disulfide bond between the catalytic domains of the E2 enzyme Ubc9 and the E1 complex subunit Uba2. In contrast, in trypanosomes, oxidative stress increased the abundance of SUMOylated protein, even at relatively low peroxide levels (32 µM, 1 h incubation; Fig. 1C). We do not know the reason for this discrepancy: the trypanosome E1 and E2 enzymes may differ such that the dimerization cannot occur, or the dimerization in mammalian cells may be caused by a specific regulatory process that is absent in trypanosomes. Peroxide concentrations above 1 mM in Saccharomyces cerevisiae (Zhou, Ryan & Zhou, 2004), or 10 mM in mammalian cells (Bossis & Melchior, 2006; Saitoh & Hinchey, 2000), increase SUMOylation, probably by inhibiting the SENP proteases (Bossis & Melchior, 2006; Xu et al., 2008). Trypanosomes probably react at lower peroxide concentrations because they are much more susceptible to oxidative stress than mammalian cells and yeast: the EC50 of hydrogen peroxide for bloodstream T. brucei is 223 µM (Krieger et al., 2000), and we found that procyclic trypanosomes were killed by concentrations above 250 µM.

Failure to purify SUMOylated proteins from T. brucei extracts

We made multiple attempts to purify the SUMOylated proteins from trypanosome extracts, using TAP-, His- and V5 tags and a variety of protocols. As previously reported for T. cruzi, all of these attempts failed (Annoura et al., 2012).

First, we attempted tandem affinity purification. SUMOylation was stable for 2 h at 4°C in the lysis buffer in which standard protease inhibitor mix and N-ethyl-maleimide (20 mM, NEM) were included in order to inhibit SUMO proteases. Although NEM was removed before the TEV protease cleavage step, we were unable to elute the tagged proteins from the column. A one-step immunoprecipitation, using V5-tagged SUMO, also yielded no specific protein pattern because only 5% of the V5-tagged SUMO bound to the beads. As SUMO binds covalently to its targets, it might be better to use His-tagged SUMO so that SUMOylated proteins can be purified under denaturating conditions.

Role of SUMOylation in T. brucei

Reciprocal BLASTp searches using yeast and human sequences, and comparison with T. cruzi (Bayona et al., 2011) revealed several putative homologues of Uba2 and Aos1, the enzymes forming the E1 complex, and also of the E2 enzyme Ubc9 (Table 2). Since SUMO E1 and E2 enzymes resemble those for ubiquitination, the specificities of these proteins is unclear. There were four possible E3 ligases, consistent with the need to regulate SUMOylation of different targets separately. However, only one SENP was found. This is surprising given that the function of SENPs include both the processing and the removal of SUMO, but the same was reported for T. cruzi (Bayona et al., 2011).

Table 2 Trypanosome genes potentially involved in SUMOylation.

Genes were identified by reciprocal BLASTp. Only genes giving a yeast SUMO pathway enzyme as the best match are included. The putative PIAS homologues each have the expected RING domain and the single SENP has a cysteine protease domain.

Enzyme	Function	Tb homologue	Name	
Aos1	E1 complex	Tb11.02.5410	AOS1	
Uba2	E1 complex	Tb927.5.3430	UBA2	
Ubc9	E2 complex	Tb927.2.2460	UBC9	
PIAS4/Siz1	E3 ligase	Tb09.211.2400		
PIAS4/Siz1	E3 ligase	Tb927.2.4420		
PIAS4/Nfi1	E3 ligase	Tb11.01.8710		
PIAS1/Siz1	E3 ligase	Tb927.6.4830		
SUMO1/Ulp2	SENP	Tb09.160.0970	SENP	

As previously reported (Obado et al., 2011), RNAi targeting SUMO in bloodstream trypanosomes halted growth 2 days after RNAi induction (Fig. 2A). We too observed numerous defects in cell division, which is normal in growth-arrested trypanosomes and does not by itself constitute evidence of a role of SUMO in regulating the cell cycle. RNAi in procyclic forms expressing V5-SUMO gave only a transient decrease in V5-SUMO (on day 2 after induction) although the RNA was clearly decreased; in two independent clones, the doubling time increased from 12.6 h to 14.5 h and 15.1 h (not shown). Liao et al. (2010) observed stronger growth inhibition.

Figure 2 (A) Effect of RNAi targeting SUMO on growth of bloodstream-form trypanosomes. RNAi was induced by addition of tetracycline and growth followed daily, with dilution as required to keep the cell density below 1 × 106/ml. (B) The effect of RNAi targeting TbUBC9 and TbSENP on SUMOylation in procyclic trypanosomes. Trypanosomes expressing T7 polymerase and the tet repressor (Alibu et al., 2004) were transfected with pHD 2021 and pHD2038 or pHD2037. RNA interference was induced with tetracycline (100 ng/ml in the absence of other selective drugs) for the times shown and the patterns of SUMOylation assayed by Western blotting.

We first targeted the putative SUMO protease, SENP (Tb09.160.0970/Tb927.9.2220). RNAi had hardly any effect on cell growth (doubling time increase of only 0.3 h, not shown), but there was a strong increase in the abundance of SUMO modification (Fig. 2B, lanes 1–4), confirming that the Tb927.9.2220 protein is important for SUMO removal in trypanosomes. Given this increase in SUMOylation, we speculate that a different enzyme might be involved in the activation of SUMO prior to transfer to the E1 conjugating enzyme. Alternatively, much lower levels of SENP activity may be needed for initial SUMO processing than for SUMO removal.

Next, we targeted the possible E2 conjugating enzyme UBC9 (Tb927.2.2460) in procyclic trypanosomes expressing V5-SUMO. A UBC9 RNAi line grew slower than the parent line, even in the absence of tetracycline, and there was only marginal slowing after tetracycline addition (not shown). We did not check the mRNA levels: the mRNA is present at less than one copy per cell so it would be difficult to detect even before RNAi (Manful, Fadda & Clayton, 2011). However, there was a reproducible decrease in SUMOylation (Fig. 2B, lanes 5–9) after RNAi.

Conclusions

We confirmed the functions of the trypanosome SENP and UBC9 genes, and could show that SUMO modifies many trypanosome proteins. The pattern of SUMOylation was surprisingly unresponsive to stress and also appeared not to be strongly developmentally regulated.

We thank Keith Gull (University of Oxford) for the anti-tubulin antibody, J Haanstra and B Bakker (Utrecht) for communicating phloretin results and Frauke Melchior (ZMBH) for advice.

Additional Information and Declarations

Competing Interests

Author Contributions

Christine Clayton is an Academic Editor for PeerJ. Otherwise, there are no competing interests.

Cornelia Andrea Klein conceived and designed the experiments, performed the experiments, analyzed the data, wrote the paper.

Dorothea Droll conceived and designed the experiments, performed the experiments, analyzed the data.

Christine Clayton conceived and designed the experiments, analyzed the data, wrote the paper.

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
