# Peer review of "SUMOylation in Trypanosoma brucei"

_PeerJ, doi:10.7717/peerj.180_

## Round 0.1 · original submission · Major Revisions

Your manuscript was reviewed by two experts and both reviewers have concerns about the presented data and request additional experiments/controls. I would re-consider a re-submission but it is critical that the major concerns identified in the critique be dealt with appropriately.

Reviewer 1 ·

Basic reporting

See below.

Experimental design

See below.

Validity of the findings

See below.

Additional comments

In this study, Klein et al. proposes that T. brucei SUMO modifies proteins and is critical for growth of bloodstream forms. The authors also suggest that SUMOylation is elevated under oxidative stress but is unchanged by temperature variation. Finally, the authors have identified two gene homologues that likely encode the E2 SUMO-conjugating enzyme (Ubc9) and the SUMO-specific protease (SENP). The paper is somewhat interesting (especially regarding the effect of oxidative stress on SUMOylation) but, disappointingly, it falls short in some key originality and experimental/methodological aspects (see below). Moreover, most of the results seem very preliminary at this point and additional experiments should be carried out, particularly concerning detection, identification, and validation of the potential SUMOylated proteins, and the role of SUMOylation in oxidative stress.

Major issues:

1) SUMOylation has recently been described in T. brucei bloodstream forms by Obaldo et al. (Nucleic Acids Res, 39:1023-33, 2011). Several of the observations regarding the role of SUMOylation in parasite cell growth currently described in this manuscript have already been reported by Obado et al. Thus, there is no major novelty on the data presented here. By the way, that key reference is surprisingly absent from the manuscript.

2) Failure in purifying and identifying any bloodstream SUMOylated protein using tandem-affinity purification (TAP) or V-5 tag. On this regard, there are many technical issues that could result in a negative TAP or V-5 purification outcome, some of them rightly pointed out by the authors: insufficient amount of protein in the initial cell lysate, low specificity of the immobilized antibody, inappropriate inhibition of SUMO proteases, low sensitivity of the methodology used for detecting the SUMOylated proteins. For instance, the authors have used SDS-PAGE trying to detect proteins after elution by TEV cleavage. It is unclear, however, what kind of gel staining they employed. This could make a huge difference in terms of sensitivity. Furthermore, very hydrophobic, high molecular mass, heavily posttranslationally modified, and/or of low abundance proteins that are not detected by SDS-PAGE (even using highly sensitive silver or fluorescence-based staining) could be identified by more sensitive approaches, such as in-solution trypsin digestion followed by liquid chromatography-tandem mass spectrometry (LC-MS/MS). Unfortunately, the authors did not explore this possibility. Also, it is unclear why the majority of TAP-tagged SUMOylated proteins could not be eluted from IgG beads. There is no discussion in that regard.

Minor issues:

1) Page 5, lines 1-2, and throughout the text, numbers with decimals are not in English format (e.g., 62,5 uM)

2) V5 Immunoprecipitation: What type of IGEPAL detergent was used? CA-630 (octylphenoxy poly(ethyleneoxy)ethanol)? There are over 30 different types of IGEPAL.

3) Please specify how exactly anti-V5 antibodies were coupled to the column? Was it by the cyanogen bromide method? The reference by Harlow and Lane (1999) is a handbook with many different coupling methods.

Reviewer 2 ·

Basic reporting

According to the abstract, the authors say that they demonstrate:
1) that SUMO modifies several proteins in T. brucei; 2) that the SUMOylation levels are not affected by changes in temperature, but are increased by oxidative stress; 3) that SUMO is required for growth of the bloodstream forms; and 4) that they obtained evidence indicating that the E2 and SENP homologs are involved in SUMOylation and the removal of SUMO, respectively.
About points 1 and 2:
That SUMO modifies a number of proteins in T. brucei has already been demonstrated by Obado et al (ref. 1 ) for the bloodstream forms.
The authors do not detect the endogenous SUMOylation pattern because of the lack of specificity of their antibodies, generated against the purified recombinant protein TbSUMO. In order to detect a SUMOylation pattern they had to do a partial replacement (of only one allele) of the endogenous gene by an N-terminal tagged form. In the case of the procyclics they use the V5 epitope and in the case of bloodstream forms the TAP tag, and then they use the corresponding commercial antibodies. The problem is, particularly when they use the 20 kDa TAP tag, that probably the tagged SUMO may not be conjugated to target proteins as efficiently as endogenous SUMO. Thus, the SUMOylation pattern presented both for normal growth conditions or under different stress conditions might not be representative of what is really occurring in the wild type parasite.
The authors do not see changes in the SUMOylation pattern either during thermal stress, or during differentiation, but they see changes during oxidative stress of the procyclic forms. However, as the authors themselves comment, in these experiments only the more abundant proteins are detected, which may not necessarily change under stress conditions; they can not discard the possibility that less abundant proteins have modified SUMOylation levels, but are not detected.
About point 3:
This has already been done and published before. The authors do not make reference to the SUMO RNAi experiments performed by Obado et al with bloodstream parasites: "The SUMO transcript was almost completely depleted within 24 h of RNAi induction (Figure 5B). However, it took at least 48 h until there was an effect on the global sumoylation pattern (Figure 5C). Western analysis of parasite lysates identified a range of sumoylated proteins, with varying intensity. The parasites were found to stop dividing after 24 h of SUMO RNAi, significant cell death was not observed until beyond the 96-h time point, nuclear division appeared to continue in the absence of cytokinesis, with many cells displaying a multi-nucleated phenotype."
About point 4) The orthologs of E2 and SENP from T. brucei were identified in silico. The authors do not demonstrate the biochemical activity of these proteins, but instead they study indirectly how the downregulation of their levels affects the global SUMOylation pattern in procyclics. The RNAi experiments are very preliminary. The authors do not show how the levels of RNA (using Northern Blot of qPCR) and protein (by Western Blot) are affected. The growth curves are not shown, and different clones are not analyzed.
In summary, this manuscript has a number of flaws in experimental design, and part of the results has little novelty, suggesting that its publication in its present form would be premature.
Ref 1:
Obado SO, Bot C, Echeverry MC, Bayona JC, Alvarez VE, Taylor MC, Kelly JM.
Centromere-associated topoisomerase activity in bloodstream form Trypanosoma brucei.
Nucleic Acids Res. 2011 Feb;39(3):1023-33

Experimental design

About the section on "Failure to purify SUMOylated proteins from T. brucei extracts":
In the case of the bloodstream forms, the reason for this failure can be a combination of factors:
- starting from a low number of parasites, as compared with the numbers of cells used for the purification of SUMOylated proteins from other organisms.
- the utilization of a rather little efficient cell breakage method for the extraction of SUMOylated proteins (since it has already been demonstrated that they are essentially nuclear proteins in Trypanosomatids).
- the insufficient inhibition of the deSUMOylases, because of the elimination of NEM during washing, or the use of a purification protocol under non-denaturing conditions.
The authors say:
"Eluates from cells expressing TAP-tagged SUMO gave exactly the same pattern on SDS-PAGE as eluates from cells expressing TAP alone and SUMO was not visible."
Was this analyzed by SDS-PAGE, staining the gels with silver? with Coomassie? The pattern observed may be identical in these cases because the SUMOylated proteins normally represent a very minor fraction of each protein target. What happens when analysis is made by Western Blot? Do they see TAP-SUMO? If they do, this would mean that most of the SUMOylated proteins were deconjugated during purification; if they do not see TAP-SUMO, this means that elution has failed.
"Further analysis indicated that the majority of TAP-tagged SUMO had bound to the IgG beads, but it was never recovered. Additional bands were however not seen after analysis of the boiled beads."
According to this statement, elution is what is not working in these experiments.
When purification from procyclics was attempted by immunoaffinity, clearly this did not work, but this does not mean that the SUMOylated proteins can not be purified.
Summarizing, the Results section entitled "Failure to purify SUMOylated proteins from T. brucei extracts" is too long and confusing. I suggest to eliminate or substantially shorten this section, saying, as Annoura et al did, that "failure of the detection of SUMOylated proteins after purification might be due to the rapid turnover of SUMO conjugation."

Validity of the findings

This manuscript has a number of flaws in experimental design, and part of the results has little novelty, suggesting that its publication in its present form would be premature.

Additional comments

No comments

---

## Round 0.2 · accepted · Accept

The author response to the critique is appropriate and the manuscript is acceptable for publication in PeerJ.